# Orthorexic Tendency in Polish Students: Exploring Association with Dietary Patterns, Body Satisfaction and Weight

**DOI:** 10.3390/nu11010100

**Published:** 2019-01-05

**Authors:** Marta Plichta, Marzena Jezewska-Zychowicz, Jerzy Gębski

**Affiliations:** Department of Organization and Consumption Economics, Faculty of Human Nutrition and Consumer Sciences, Warsaw University of Life Sciences (SGGW-WULS), 159C Nowoursynowska Street, 02-787 Warsaw, Poland; marzena_jezewska_zychowicz@sggw.pl (M.J.-Z.); jerzy_gebski@sggw.pl (J.G.)

**Keywords:** dietary patterns, body satisfaction, orthorexia nervosa, students

## Abstract

Body dissatisfaction is central to clinically diagnosed eating disorders (ED) and seems to be important in causing other non-clinical disorders, including orthorexia nervosa (ON). It can also affect eating behaviors. The aim of this study was to assess the associations of ON tendency with dietary patterns (DPs) and body satisfaction. The data were collected in 2017 through questionnaire survey among 1120 students of health-oriented and other academic programs from seven universities in Poland. Principal components analysis (PCA) was conducted to derive DPs and body satisfaction factors. Six DPs, such as, ‘High-sugar products & snacks’, ‘Fresh products & nuts’, ‘Fatty products & dressings’, ‘Oils & potatoes’, ‘Dairy products & whole-meal bread’, ‘Meat’, and two body satisfaction factors, such as, ‘Bottom body & weight’, and ‘Upper body’ were identified. ON tendency was measured using ORTO-15 questionnaire with both cut-offs, i.e., 35 and 40. Logistic regression analysis was used to verify associations between ON tendency, body satisfaction factors, and DPs. More students of health related majors were characterized by ON tendency in comparison to students of other majors (35.9 vs. 37.2; *p* < 0.001). More women were dissatisfied with ‘Bottom body & weight compared to men (<0.001). The higher the body mass index (BMI), the more people were dissatisfied with ‘Bottom body & weight’ (*p* < 0.001). More students with ON tendency were satisfied with their ‘Upper body’ than those without ON tendency, but there were no differences in ON tendency in regard to ‘Bottom body & weight’ satisfaction. ON tendency was associated with more frequent consumption of vegetables, fruits, nuts and seeds, and less frequent consumption of products high in sugar, snacks, fatty products and dressings. Using cut-off at 35 in ORTO-15 seems to be more appropriate than cut-off at 40 to identify external variables describing ON. Future research on orthorexia nervosa should use other research tools than ORTO-15 to better identify individuals with ON and to confirm our findings.

## 1. Introduction

Dissatisfaction with one’s body, distorted body image perception and obsession with thinness are central to clinically diagnosed eating disorders (ED) such as anorexia nervosa (AN), bulimia nervosa (BN), binge eating disorder (BED) and eating disorders not otherwise specified (EDNOS) [1]. It can be assumed, that these factors may also contribute to other non-clinical types of disorders including orthorexia nervosa (ON). ON expresses a pathological interest in healthy food. The healthy behavior and ON are placed as extreme poles on the continuum scale of states differing in the intensity of orthorexic symptoms [2].

ON was used to define an unhealthy obsession with healthy eating [3]. Individuals with ON eliminate certain food products that are perceived by them as unhealthy and impure. They avoid foods with high content of salt, sugar and fat, foods containing genetically-modified ingredients, herbicides, pesticides and artificial substances and also non-organic foods. Each meal is prepared with great concern and attention, often with repetitive activities which are to ensure the safety of food intake. This obsession is driven by a desire to optimize own health and well-being [4]. Individuals with ON may be on alternative diets (e.g., vegetarianism, veganism, raw foods, macrobiotics, fruitarianism) [5,6,7]. Recently, Dunn and Bratman [8] proposed the new criteria of classification of ON containing extreme preoccupation with healthy eating, restrictive eating behaviors, malnutrition, severe weight loss as a results of food choices as well as feeling guilt or self-loathing if diet is not followed correctly. However, no standardized tool for diagnosis currently exists [9,10,11,12,13].

To assess ON, ORTO-15 questionnaire [14] based on Bratman Orthorexia Test (BOT) [4] is widely used, despite criticism regarding this research tool [7]. Some researchers suggested that ORTO-15 questionnaire may not be able to distinguish pathological behaviors and is not clinically relevant [15]. However, the recent review of the studies using ORTO-15 showed that Cronbach’s alpha coefficients were ranging from 0.83 to 0.91 [16]. The Italian version of ORTO-15 was translated into several languages, usually with the use of complicated multistep methods, and then administered to different samples. The results of confirmatory analysis and other methods allowed researchers to shorten original ORTO-15 [7]. In this way, the Turkish version of ORTO-11 [10] and ORTO-11-Hu, Hungarian adaptation of the ORTO-15 [12], were developed. In the original version of ORTO-15, to distinguish the orthorexic tendency it is recommended to use the cut-off point below 40 [17], and some researchers followed this recommendation [18,19]. Nevertheless, there are also suggestions to lower the cut-off point in this test to 35 [11,18]. Lowering the cut-off point would allow distinguishing a group of people in which the ON symptoms can be clearly identified. Thus, it enables a more effective search for the ON symptoms, its causes and consequences.

ON reveals several similarities with other eating disorders, such as avoidant/restrictive food intake disorder (AFRID) [20] and AN [21]. AFRID is characterized by the lack of interest in food, avoiding food products with specific shapes or colors, or anxiety about the aversive consequences of eating [22]. Nevertheless, the anxiety about eating results from a response to a preceding traumatic event (e.g., choking) or an aversive experience (e.g., repeated vomiting) [22,23] and not from excessive concerns for health as in ON. In contrast, AN and ON share traits of perfectionism, cognitive rigidity, high co-morbid anxiety, and need for control of life transferred to eating [5,21]. Both patients with AN and patients with ON perceive their behaviors as ego-syntonic [24]. Nonetheless, patients with ON are concerned about food quality rather than food quantity in contrast to patients with AN [25]. In addition to AN characteristics, individuals with ON may evince obsessive-compulsive disorders (OCD) [26] due to burdensome obsessions (e.g., intensive meal planning, intrusive thoughts of food) and compulsions (e.g., exact measuring and weighing of food). However, the obsessions in OCD are described as ego-dystonic [6].

ON may lead to severe physical and psychological complications such as, caloric deficit, lack of vitamins and minerals, gastrointestinal problems, stomach inflammation, anemia, osteopenia, hyponatremia, pancytopenia, heart failure, significant distress, and social isolation [2,5,27,28]. However, ON is still not diagnosed well and not recognized in the Diagnostic and Statistical Manual of Mental Disorders DSM-5 [1]. Thus, further research is needed to better understand the symptoms of ON [27,29]. 

Despite certain similarities between AN and ON, some researchers suppose that ON is not associated with weight control [24,30]. Nevertheless, young people are often dissatisfied with their bodies [31,32], and such negative body image may put them at severe health risk because it strongly predicts the development of disordered eating [33]. It has been showed that females with ON were characterized by striving for thinness, lower body satisfaction, and lower body acceptance [34]. Previous studies have shown that increased ON symptomatology was associated with calorie restriction and preoccupation with weight [10,13,18,35,36,37,38,39]. Nevertheless, the relationship of ON and body satisfaction requires further research. 

Orthorexic individuals represented very specific eating habits, among others they avoid foods that may include genetically modified ingredients or pesticides, foods high in salt or sugar, and foods high in fat [24,40]. Previous research has shown that individuals eating whole wheat cereals, as well as fruits and vegetables more frequently were characterized by higher ON tendency compared to others [41]. However, despite the fact that ON is characterized by very specific eating behaviors, still little is known about food intake frequency in relation to this disorder [12,41]. To the best of our knowledge, the relationship between dietary patterns (DPs), body satisfaction and ON tendency was not examined. Therefore the aim of our study was first, to estimate the orthorexic tendency among students of different college majors, and second, to find the associations between ON tendency, body satisfaction, and dietary patterns in students taking into account gender, body mass index (BMI) and college majors, and third to recognize the usefulness of both cut-offs used to identify orthorexic tendency (at 35, and 40 points in ORTO-15) in the determination of external variables describing ON symptoms including dietary patterns and body satisfaction.

## 2. Materials and Methods

### 2.1. Ethical Approval

The study protocol was approved by the Ethics Committee of Faculty of Human Nutrition and Consumer Science, Warsaw University of Life Sciences (Resolution number 45/2017). Informed consent to participate in the study was collected from all participants.

### 2.2. Participants and Procedure

The study sample was selected in seven universities from Poland. The survey was conducted during lectures by a trained person. The teacher left the room after consenting to the survey among students. The persons participating in the study were informed about the purpose, course and duration of the study, as well as about the possibility of resigning from the study at any stage without bearing any legal liability. Participation in the study was voluntary. The questionnaire took approximately 15 min to complete. In total, 1300 students of health-oriented and other academic programs were invited to participate in the study. Male and female students at the age from 18 to 35 years old were included in the sample. As a result of incomplete questionnaires, 162 participants were excluded from the sample. Moreover, 18 students were excluded from the sample as they were older than 35 years old. Finally, the sample consisted of 1120 students who completed questionnaires correctly. The data were collected in 2017.

### 2.3. Questionnaire

The ORTO-15 questionnaire [19] was used to identify the ON tendency. The Polish version used in our study was adapted from the original Italian ORTO-15 [14], and has good repeatability and satisfactory reliability (Cronbach’s alpha 0.70–0.90) [19]. The ORTO-15 is a validated measure consisting of 15 items that assesses beliefs about the perceived effects of eating healthy food, attitudes conditioning food selection, habits regarding food consumption, and the extent to which food concerns affect daily life [24]. Responses expressing the obsessive attitude of the individuals in choosing, buying, preparing and consuming food are scored on a 4-point scale (always, often, sometimes, never). Responses indicating ON symptoms were scored 1, while responses describing adequate eating behaviors were scored 4. The sum of scores was calculated for each participant with a minimum of 15 and a maximum of 60. 

The Questionnaire of Body Particular Parts and Parameters Satisfaction [42] was used to measure body satisfaction. This tool was developed and then validated in the Polish population. The obtained reliability index for the entire scale was considered satisfactory (Cronbach’s alpha 0.86) [42]. Respondents defined the satisfaction level with particular body parts on a nine-point scale, where: 1—totally dissatisfied; 5—neither dissatisfied nor satisfied; 9—totally satisfied. The following parts of the body were included in the questionnaire: face; shoulders; breast/chest; waist/midsection; abdomen; hips; thighs; legs; weight and height of body.

The Food Frequency Questionnaire (FFQ-6) was applied to determine respondents’ dietary habits [43]. The FFQ-6 was validated in females aged 13–21 years old [43] and in both men and women aged 21–26 years old [44]. The FFQ-6 covered the diet thoroughly, including many of the most popular dishes consumed in Poland. The FFQ contained a list of 165 products and dishes which represented such food groups as: sweets and snacks; dairy products and eggs; cereal products; fats; fruits; vegetables and grains; meat and fish products; beverages. The frequency of food consumption throughout the year preceding the study was assessed. For this analysis, 26 products and dishes were chosen, including: sugar; chocolate, chocolate candies and chocolate bars; non-chocolate candies; biscuits and cookies; salty snacks; milk and natural milk drinks; sweetened milk drinks; cheese; whole-meal bread; refined bread; oils; butter; margarine; sour cream and sweet cream; other animal fats; mayonnaise and dressings; fresh fruits; processed fruit products; fresh vegetables; potatoes; nuts; grains and wheat germ; red meat; poultry meat and rabbit meat; fatty fish; sweetened fizzy drinks. When selecting 26 products, the results of other studies on the relationship between body dissatisfaction and consumed foods were taken into account [45,46]. In addition, the choice of items resulted from their importance in diet of individuals with ON tendency. FFQ-6 included six categories of food intake frequency: never or almost never (1 points); once a month or less frequently (2 points); several times per month (3 points); several times per week (4 points); every day (5 points); and several times per day (6 points). 

Socioeconomic profile of participants was assessed based on questions regarding gender, age (in years), place of residence, and college major. Body Mass Index (BMI) was calculated using self-reported weight and height and categorized accordingly World Health Organization [47] into four groups: underweight (BMI < 18.5 kg/m^2^), normal weight (18.5 ≤ BMI ≤ 24.9 kg/m^2^), overweight (25.0 ≤ BMI ≤ 29.9 kg/m^2^) and obesity (BMI ≥ 30.0 kg/m^2^).

### 2.4. The Sample Characteristics

The characteristics of the sample are presented in Table 1. The sample included 1120 college students, of which 70.4% were female and 29.6% were male. The highest number of participants were younger than 20 years old (36.1%). Majority of the students lived in cities (71.6%). More participants were enrolled in bachelor or engineering studies (88.7%) than in master studies (11.3%). Almost a half of the sample (48.8%) represented health related majors (Food Technology and Human Nutrition, Dietetics, Physiotherapy, Physical Education and Wellness) and 51.2% were students of other majors. The majority of participants had normal weight (72.9%).

### 2.5. Data Analysis

Descriptive statistics including frequency distributions and cross-tabulations were carried out. The Principal Component Analysis (PCA) was applied to identify body satisfaction factors and dietary patterns (DPs), separately. The factors were orthogonally rotated (the varimax option) to maintain uncorrelated factor variables. The number of factors was based on the following criteria: components with an eigenvalue of 0.1, scree plot test and the interpretability of the factors. The factorability of data was confirmed with Kaiser–Meyer–Olkin (KMO) measure of sampling adequacy and Bartlett’s test of sphericity achieving statistical significance. Bartlett’s tests have a significance of *p* < 0.001 for both data. KMO value for body satisfaction data was 0.862 and for dietary data was 0.842. Factor loadings higher than 0.4 assure that the variables extracted are shown through a specific factor, therefore both food items and body parts with factor-loadings of at least 0.50 have been taken into consideration and used to label the dietary patterns and body satisfaction factors, respectively [48]. 

Two body satisfaction factors were identified: ‘Bottom body & weight’ and ‘Upper body’. Total variance explained was 57.6%. The factor-loading matrix for the body satisfaction identified by principal component analysis (PCA) is presented in detail in Table 2.

Six dietary patterns were identified: ‘High-sugar products & snacks’, ‘Fresh products & nuts’, ‘Fatty products & dressings’, ‘Oils & potatoes’, ‘Dairy products & whole-meal bread’, and ‘Meat’. Total variance explained was 51.3%. For each factor the explained variance was: 19.1%, 10.9%, 6.7%, 5.4%, 4.8%, and 4.4%, respectively. The factor-loading matrix for the DPs identified by principal component analysis (PCA) is presented in detail in Table 3.

Respondents allocated to each of the dietary patterns were divided into three groups based on tertile distribution: bottom, middle and upper tertile. Participants in the upper tertile had the strongest adherence to the pattern.

Two categories within each body satisfaction factors, such as, satisfaction and dissatisfaction, were created using factor scores obtained with regression approach. The factor scores of 0 and below is interpreted as ‘dissatisfaction’, whereas the value above 0 as ‘satisfaction’ with body.

According to Donini et al. [14], a score lower than or equal to 40 indicates ON tendency, while higher scores inform about adequate eating behaviors. Whereas according to Ramacciotti et al. [11], a cut-off at 35 provides a lower and more accurate estimate of ON tendency, than the cut-off at 40. Both cut-offs were included in our study. Cronbach’s alpha for ORTO-15 was 0.7.

The differences between groups were verified by Chi-square test (the ON tendency groups versus sociodemographic characteristics, including gender, age bracket, place of residence, level of studies, college major and BMI categories); student’s *t*-test (mean score of ORTO-15 versus gender and college majors); the one-way analysis of variance ANOVA (mean score of ORTO-15 versus BMI categories).

A simple logistic regression analysis was applied to the models verifying associations between ON tendency, body satisfaction, and DPs. The odds ratio (OR) and 95% confidence interval (95% CI) were calculated. Such dichotomous variables as ON tendency, ‘Bottom body & weight’ and ‘Upper body’ have been introduced into models as dependent variables. The independent variables in the models were dietary patterns. The OR calculated for each independent variable represents the odds ratio between a given variable level (the upper tertile of each dietary pattern), and the adopted level of reference (the bottom tertile—OR = 1.00) while the remaining explanatory (independent) variables remain constant. The adjusted models were developed for the potential confounders, such as, gender (categorical, female/men), age (continuous, years), place of residence (categorical, village/city ≤ 100.000 citizens/city > 100.000 citizens), level of studies (categorical, bachelor or engineering studies/master studies), college major (categorical, health related/non health related), and BMI (continuous, kg/m^2^). The level of significance of the odds ratio was assessed with a Wald test. A *p*-value ≤ 0.05 was considered statistically significant for all tests. All analyses were carried out applying sample weights to adjust for non-response and missing data. All analyses were performed using SAS 9.4. software (SAS Institute, Cary, NC, USA).

## 3. Results

### 3.1. Orthorexia Nervosa Tendency and Body Satisfaction across Gender, BMI and College Major

The ON tendency was observed among 75.0% of the sample when the cut-off at 40 was used. However, only 28.3% of students showed ON tendency when the cut-off at 35 was implemented. Mean value of ORTO-15 was 36.6 (standard deviation 4.2). More than a half of participants were satisfied with both ‘Bottom body & weight’ (52.9%) and ‘Upper body’ (51.4%). Twice more females than males were dissatisfied with ‘Bottom body & weight’ (*p* < 0.001). There were no significant differences in ON tendency and ‘Upper body’ satisfaction between gender groups (Table 4).

The differences between BMI categories regarding ‘Bottom body & weight’ satisfaction were observed (<0.001). The higher the BMI, the more people were dissatisfied with ‘Bottom body & weight’. No significant differences were observed between BMI categories in regard to the ON tendency (regardless of the cut-off point) and ‘Upper body’ satisfaction (Table 5). 

Students of health related majors (35.9 ± 4.1) scored significantly lower in ORTO-15 than students of other majors (37.2 ± 4.2). More students of health related majors (79.3%) than students of other majors (70.9%) displayed ON tendency (cut-off at 40). Similarly, such difference was observed with the cut-off at 30 (32.9 and 23.9%, respectively). There were no significant differences both in ‘Bottom body & weight’ and ‘Upper body’ satisfaction across the college majors (Table 6).

### 3.2. Orthorexia Nervosa Tendency and Body Satisfaction

More students with ON tendency were satisfied with their ‘Upper body’ in comparison with students without ON tendency. There was not a significant difference in ‘Bottom body & weight’ satisfaction in respect to ON tendency. Such relationships were demonstrated for categories identified using both cut-offs (*p* = 0.027 and *p* = 0.005, respectively at 40 and 35) (Table 7).

### 3.3. Associations between Orthorexia Nervosa Tendency, Body Satisfaction and Dietary Patterns

The results from the adjusted model have demonstrated that students who consumed high-sugar products and snacks most often (the upper tertile of DP) were less likely to display ON tendency, regardless of the cut-off point. Odds Ratio (OR) was higher at the cut-off at 35 points compared to the cut-off at 40 points (0.32 and 0.34, respectively). In addition, students in the upper tertile of ‘Fatty products & dressings’ pattern were less likely to display ON tendency compared to those in the bottom tertile of this DP. However, Odds Ratio (OR) was lower at the cut-off at 35 compared to the cut-off at 40 (0.52 and 0.59, respectively). In contrast, students in the upper tertile of ‘Fresh products & nuts’ were more than twice as likely to have ON tendency compared to those in the bottom tertile of this DP. Odds Ratio (OR) was higher at the cut-off at 35 compared to the cut-off at 40 (2.49 and 2.30, respectively) (Table 8). 

Students in the upper tertile of ‘High-sugar products & snacks’ were more likely to be dissatisfied with their ‘Upper body’ ((OR) 1.75, 95% CI: 1.29–2.37) compared to those in the bottom tertile. Students who consumed fresh products and nuts most often (the upper tertile of ‘Fresh products & nuts’) were less likely to be dissatisfied with their ‘Upper body’ ((OR) 0.65, 95% CI: 0.48–0.88) than those in the bottom tertile. No significant associations between ‘Bottom body & weight’ satisfaction and DPs were revealed (Table 8).

## 4. Discussion

In our study we attempted to assess the occurrence of the ON tendency amongst students, who, due to their young age, are prone to developing incorrect eating behaviors [49]. Our results obtained in the ORTO-15 test for the cut-off of 40 are in accordance with other studies [12,38,50,51]. After reducing the cut-off limit to 35 points, there was almost a three-fold decrease in the number of people with ON tendency (from 75.0% to 28.3%). Other researchers also pointed out the validity of lowering the cut-off below 35 points in the ORTO-15 test [11,18].

Previous research indicate that the occurrence of ON is more common among students of medicine [10], exercise science [52], dietetics [53] and majors related to nutrition [9]. Our results also confirm these observations in relation to health-related majors. Both the percentage of people with ON symptoms and the mean value of ORTO-15 indicate differences between students of majors related to health and the rest of participants, but these differences were not of clinical significance. However, they can be used in the development of education programs. Students of health-related majors acquire knowledge about the principles of adequate nutrition and health consequences resulting from their non-observance. In addition, they use other sources of information (e.g., Internet sources) about food and its impact on health [54]. This may result in the application of strict nutritional rules aimed at maintaining health or normal body weight, however this can also lead to the development of eating disorders [55,56]. 

Our study did not reveal differences in the occurrence of ON tendency between the categories identified due to BMI, both for the cut-off limit at 35, and at 40 points in the ORTO-15 test. Previous research did not confirm the association between ON and BMI as well [36,57,58], which indicates the distinctiveness of ON from AN in terms of this index [21,24,59,60]. Excessive concentration of people with the ON tendency on the quality of food [24] does not have to be combined with control of the amount of food consumed. In studies on stereotypical thinking about healthy food, it was shown that when people assessed its impact on weight gain mainly the health-related properties of the food consumed were taken into account and not its quantity [61,62]. The lack of association between orthorexic symptoms and BMI, but also stereotypical thinking of healthy food as good for body could therefore suggest that encouraging the choice of healthy food is not sufficient to limit overweight and obesity. The recommendation to choose healthy food has to be supplemented with the recommendation to control the amount in which healthy food is consumed.

The results of our study revealed the relationship of BMI with the student’s satisfaction with ‘Bottom body & weight’, with the highest number of underweight individuals being satisfied with ‘Bottom body & weight’. Likewise Goswami et al. [63] we observed a higher level of body satisfaction among the respondents with BMI < 18.5 kg/m^2^ in comparison to people with higher BMI (>25 kg/m^2^). The majority of obese people were dissatisfied with their ‘Bottom body & weight’, which may result from excess of body fat deposited especially in the lower body parts, as well as excessive body weight. The overweight or obese individuals may experience strong stigma and discrimination due to weight and size of their body, leading them to focus increasingly and obsessively on dieting for weight loss and body dissatisfaction [64]. Low body satisfaction may lead to development of an ED, as well as to sadness, low self-esteem and depression [49]. However, it is alarming that among people with a normal body weight, almost half were dissatisfied with ‘Bottom body & weight’, which may lead them to implementing dietary restrictions. An important factor causing the lack of satisfaction with one’s own body is the internalization of information provided by parents, peers and mass media [65]. Interestingly, the messages addressed to men and women significantly differ from each other [66]. In the case of women, the information transmit the pressure of having a slim figure. For men, the pressure concentrates on having an athletic and muscular body. Dissatisfaction with one’s own body may stem from the inability to meet the imposed ideals [67]. Women internalize messages regarding the body image more often and they tend to be more severe judges of their looks in comparison to men [68,69,70]. This tendency occurs alike among children [71], adolescents [72], and grown-ups [73]. The lack of associations between body satisfaction and BMI demonstrated in our study can be used in the development of messages aimed at reducing obesity. In order to encourage people to maintain or reduce weight, both health and body satisfaction should be emphasized as resulting from controlling one’s diet. Body satisfaction would be presented as a result of the action undertaken, and at the same time a reward for the effort made, which may increase the effectiveness of the process of behaviors’ change.

Although more women than men experience ED [74,75,76,77], our results did not confirm the relationship between gender and ON tendency regardless of the cut-off point used in the ORTO-15 test. Previous studies also did not show any definite relationship between gender and ON. Some research papers have pointed to a higher incidence of ON among men [10,14,78], whereas others observed it more often in women [26,36,79]. Considerable number of publications, like our study, did not indicate any significant relationship between gender and ON [80,81]. The discrepancies in results may be related to the specificity of ON. An unhealthy obsession with proper nutrition resulting from the preoccupation with one’s health is the main characteristic of ON [3,4]. The increasing awareness of the relationship between nutrition and health induces growing interest in this issue [82], which may favor the application of strict nutrition rules regardless of gender. Their use can reach the pathological level and turn into ON, also regardless of gender. In the case of ED, such as AN or BN, dissatisfaction with body image, fear of obesity and distorted assessment of body dimensions are the main factors affecting the formation of these disorders [1]. Thus, women who are generally less satisfied with the image of their own body constitute a group that is particularly vulnerable to this type of disorders. 

Previous studies did not indicate a connection between the negative image of one’s body and ON [30,83], but the fact that the perception of one’s own body is a frequent cause of changing one’s eating behaviors has prompted us to include this variable in our study. The obtained results indicate the lack of relationship between ON tendency (both cut-offs in ORTO-15 test) and satisfaction with own body, but only in regard to ‘Bottom body & weight’. In ED, concerns about the body image refer primarily to those parts of the body (i.e., abdomen, hip, thighs and buttocks) in which fatty tissue accumulates [42]. The lack of this relationship could confirm the differences between ON and BN, due to the lack of a characteristic symptom of AN and BN (i.e., body dissatisfaction). Nevertheless, our results indicate a relationship of satisfaction with the upper body with ON tendency. Higher satisfaction with the upper body of people with ON tendency may result from the fact that they eliminate products with a high sugar content and processed food from their diet [3,4], which may affect skin condition adversely [84]. Our results confirm that the respondents from the upper tertile of ‘High-sugar products & snacks’ pattern were more likely to be dissatisfied with upper body and those in the upper tertile of ‘Fresh products & nuts’ pattern were less likely to be dissatisfied with upper body. Thus, the restrictions introduced by people with ON tendency can positively affect the skin condition of these parts of the body, and thus increase the satisfaction with their body. 

Our research has pointed to the lower risk of ON tendency (regardless of the cut-off limit in ORTO-15) among the respondents in the upper tertile of ‘High-sugar products & snacks’ and ‘Fatty products & dressings’ patterns. People with ON avoid foods rich in sugar, fat and salt [13,24,41], which is confirmed by our results. In addition, in our study greater ON tendency (for both cut-offs in ORTO-15 test) in people from the upper tertile of ‘Fresh products & nuts’ pattern was observed. An earlier study also showed higher intake of vegetables and fruits among people with ON [12]. Our results did not confirm the relationship between ‘Meat’ DP and ON tendency, regardless of the cut-offs used in ORTO-15. Some researchers suggest that vegetarian diet does not lead directly to ED, however, the prevalence of ON among vegans and vegetarians is higher than among people who eat meat [85,86,87,88]. It seems that only the complete elimination of meat from the diet, and not the varied frequency of its consumption may be the reason for the more frequent occurrence of ON tendency among vegans and vegetarians. However, further research into ON should be carried out in the context of the use of alternative diets.

The use of both cut-off points in our analysis (at 35 and at 40 points in ORTO-15) in the determination of ON tendency has not provided us with findings which we expected. There were no major differences in the results after applying both cut-offs, except for the estimation of the occurrence of ON tendency. The differences shown in the dietary patterns after taking into account both cut-off points in ORTO-15 were only minimal. However, slightly higher chances of representing the upper tertile of ‘Fresh products & nuts’ and lower chances of representing the upper tertile of ‘Fatty products & dressing’ were observed in people with higher symptoms of ON when the cut-off at 35 was used, which is also confirmed in other studies [12,13,24,41]. Although the use of cut-off at 35 in ORTO-15 seems more appropriate to diagnose external variables describing ON, further studies are needed to confirm our results. 

### Strengths and Limitations

The strength of our results is a relatively large sample of Polish students. Although our findings are specific to Polish cultural background and should not be generalized to populations of students of various nationalities, the findings could be of potential use in further research on ON tendency and its symptoms, especially when methodological issues are concerned. We believe that including body satisfaction and dietary patterns in the study brought a wider perspective on symptoms of ON tendency.

Nevertheless, there are several limitations within the study. The first limitation of the current study is carrying out the survey in lectures, which limited the anonymity of the respondents. The literature suggests that anonymity may enhance the accuracy of disclosure among college students for questions of a sensitive nature [89]. Secondly, the self-reported weight and height could have led to an inaccurate body mass index classification. The ability to self-report weight and height data may be influenced by sociodemographic features such as age, gender and economic status [90]. Next, the three identified dietary patterns explained only 5.4% or less of the total variance in the dietary intake data, which is lower compared to another studies undertaking similar type of analyses [91,92]. This value may be a results of the inclusion of a high numbers of variables in the PCA [93]. Finally, the cross-sectional design of the present study and data collection at a single point in time did not allow conclusions to be drawn about causality, but only on the associations of ON tendency with both dietary patterns and body satisfaction. Moreover, using the ORTO-15 for examining ON tendencies can be questioned due to its limitations [94,95].

## 5. Conclusions

The results of our study indicate that there was no body dissatisfaction among people with ON tendency. What’s more, people with ON tendency were more satisfied with their upper body in comparison to people with no orthorexic characteristics. In addition, there was no relationship between ON tendency and BMI. The obtained results do not confirm the suggested relationship between ON and AN due to the lack of ON relationship with dissatisfaction with one’s body and BMI. However, further research in this area is needed in order to confirm the relationship between the one’s body satisfaction and ON in different populations, including youth that does not attend college, adults and people with ED. Moreover, the assessment of the relationship of ON tendency with dietary patterns allowed characterizing dietary behaviors of people with ON tendency, which may be important in the development of educational programs and nutritional interventions directed to people with ON.

The use of both cut-off points for results obtained in ORTO-15, and then their inclusion in the analyses suggests the legitimacy of using the cut-off point 35, because it results in a decrease in the number of people with ON tendency. However, in the diagnosis of ON symptoms such as body satisfaction and dietary patterns, there were no major differences from the cut-off point at 40. This can confirm the limitations of the ORTO-15 demonstrated by other researchers. Future research on ON should use other research tools than ORTO-15 to identify individuals with ON and to confirm our findings. Simultaneously, research should be intensified in order to create a new tool for better identification of ON and its symptoms, both behavioral and related to psychological characteristics.

## Figures and Tables

**Table 1 nutrients-11-00100-t001:** Sociodemographic characteristics of the study sample.

Variables		*N* = 1120	%
Gender	Female	789	70.4
	Male	331	29.6
Age	18–19	404	36.1
	20–22	388	34.6
	23–25	253	22.6
	26–35	75	6.7
Place of residence	Village	318	28.4
	City ≤ 100.000 citizens	319	28.5
	City > 100.000 citizens	483	43.1
Level of studies	Bachelor or engineering studies	994	88.7
	Master studies	126	11.3
College major	Health related	547	48.8
	Non health related	573	51.2
BMI categories	Underweight	123	11.0
	Normal weight	817	72.9
	Overweight	154	13.8
	Obesity	26	2.3

*N*—number of participants, BMI—body mass index.

**Table 2 nutrients-11-00100-t002:** Factor-loading matrix for the body satisfaction factors identified by principal component analysis (PCA).

Variables	Bottom Body & Weight	Upper Body
Waist/midsection	**0.617**	0.438
Abdomen	**0.692**	0.321
Hips	**0.800**	0.261
Thighs	**0.853**	0.137
Legs	**0.795**	0.141
Weight	**0.758**	0.436
Face	0.186	**0.715**
Shoulders	0.282	**0.666**
Breasts/chest	0.064	**0.750**
Variance Explained (%)	46.0	11.6
Total Variance Explained (%)	57.6	
Kaiser’s Measure of Sampling Adequacy	0.862	

Bolded values are marked for the main components of PCA-derived body satisfaction factors with absolute loadings ≥ 0.5.

**Table 3 nutrients-11-00100-t003:** Factor-loading matrix for the DPs identified by principal component analysis (PCA).

Variables	High-Sugar Products & Snacks	Fresh Products & Nuts	Fatty Products & Dressings	Oils & Potatoes	Dairy Products & Whole-meal Bread	Meat
Chocolate, chocolate candies and chocolate bars	**0.722**	−0.094	−0.020	0.090	0.084	0.016
Biscuits and cookies	**0.709**	−0.023	0.028	0.115	0.195	0.032
Non chocolate candies	**0.757**	0.024	0.103	−0.046	0.006	0.125
Salty snacks Sweetened milk drinks	**0.580** **0.506**	−0.093 −0.083	0.251 0.243	0.121 −0.017	−0.075 0.461	0.045 0.022
Milk and natural milk drinks	0.095	0.073	−0.001	0.007	**0.785**	0.034
Cheese, blue cheese, melted cheese, spread cheese	0.111	−0.075	0.250	0.159	**0.616**	0.034
Whole-Meal bread	−0.085	0.249	−0.225	0.054	**0.525**	−0.058
Margarine Sour cream and sweet cream Other animal fats Mayonnaise and dressings	0.190 0.178 0.048 0.229	−0.028 −0.001 0.139 −0.066	**0.561** **0.537** **0.640** **0.502**	0.064 0.312 0.047 0.298	0.012 0.312 −0.051 0.166	−0.187 0.073 0.381 0.141
Fresh fruits Fresh vegetables	−0.006 −0.207	**0.640** **0.594**	−0.372 −0.328	0.304 0.311	0.134 0.106	−0.088 0.013
Nuts	−0.099	**0.728**	0.084	−0.058	−0.037	0.093
Seeds (e.g., pumpkin, sesame, sunflower, wheat germ)	−0.068	**0.738**	0.104	−0.111	0.018	−0.047
Red meat (e.g., pork, beef, veal)	0.065	−0.010	0.233	0.139	−0.018	**0.770**
Poultry meat and rabbit meat	0.115	−0.039	−0.095	0.123	0.030	**0.764**
Oils	−0.008	0.172	−0.008	0.658	0.082	0.068
Potatoes	0.258	−0.017	0.159	0.631	−0.053	0.129
Variance Explained (%)	19.1	10.9	6.7	5.4	4.8	4.4
Total Variance Explained (%)	51.3					
Kaiser’s Measure of Sampling Adequacy	0.842					

Bolded values are marked for the main components of PCA-derived dietary patterns factors with absolute loadings ≥ 0.5.

**Table 4 nutrients-11-00100-t004:** Orthorexia nervosa tendency and body satisfaction in the total sample and gender groups.

Variables	Total Sample	Female	Male	Chi-Squared Test/Student’s *t*-Test	*p*-Value
	*N* = 1120	*N* = 789	*N* = 331		
Orthorexia nervosa *n* (%)					
ON tendency (<40)	840 (75.0)	595 (75.4)	245 (74.0)	0.24 *	0.623
Without ON tendency (<40)	280 (25.0)	194 (24.6)	86 (26.0)		
ON tendency (<35)	317 (28.3)	219 (27.8)	98 (29.6)	0.39 *	0.530
Without ON tendency (<35)	803 (71.7)	570 (72.2)	233 (70.4)		
ORTO-15 score (M ± SD)	36.6 ± 4.2	36.6 ± 4.1	36.6 ± 4.4	0.03 **	0.973
‘Bottom body & weight’ *n* (%)					
Dissatisfaction	527 (47.1)	437 (55.4)	90 (27.2)	74.41 *	<0.001
Satisfaction	593 (52.9)	352 (44.6)	241 (72.8)		
‘Upper body’ *n* (%)					
Dissatisfaction	544 (48.6)	372 (47.2)	172 (52.0)	2.16 *	0.141
Satisfaction	576 (51.4)	417 (52.8	159 (58.0)		

* Chi-squared test; ** Student’s *t*-test (t); ON—orthorexia nervosa; M—mean; SD—standard deviation.

**Table 5 nutrients-11-00100-t005:** Orthorexia nervosa tendency and body satisfaction according to BMI categories.

Variables	Underweight	Normal Weight	Overweight	Obesity	Chi-Squared Test/ANOVA	*p*-Value
	*N* = 123	*N* = 817	*N* = 154	*N* = 26		
Orthorexia nervosa *n* (%)						
ON tendency (<40)	91 (74.0)	620 (75.9)	111 (72.1)	18 (69.2)	1.57 *	0.665
Without ON tendency (<40)	32 (26.0)	197 (24.1)	43 (27.9)	8 (30.8)		
ON tendency (<35)	38 (30.9)	230 (28.2)	43 (27.9)	6 (23.1)	0.78 *	0.855
Without ON tendency (<35)	85 (69.1)	587 (71.9)	111 (72.1)	20 (76.9)		
ORTO-15 score (M ± SD)	36.5 ± 4.4	36.6 ± 4.1	36.6 ± 4.6	37.7 ± 3.6	1.23 **	0.197
‘Bottom body & weight’ *n* (%)						
Dissatisfaction	32 (26.0)	382 (46.8)	93 (60.4)	20 (76.9)	42.18 *	<0.001
Satisfaction	91 (74.0)	435 (53.2)	61 (39.6)	6 (23.1)		
‘Upper body’ *n* (%)						
Dissatisfaction	70 (56.9)	389 (47.6)	69 (44.8)	16 (61.5)	6.35 *	0.096
Satisfaction	53 (43.1)	428 (52.4)	85 (55.2)	10 (38.5)		

* Chi squared test; ** one-way analysis of variance ANOVA (F).

**Table 6 nutrients-11-00100-t006:** Orthorexia nervosa tendency and body satisfaction according to majors of the study.

Variables	Health Related Majors	Non Health Related Majors	Chi-Squared Test/Student’s *t*-Test	*p*-Value
	*N* = 547	*N* = 573		
Orthorexia nervosa *n* (%)				
ON tendency (<40)	434 (79.3)	406 (70.9)	10.75 *	0.001
Without ON tendency (<40)	113 (20.7)	167 (29.1)		
ON tendency (<35)	180 (32.9)	137 (23.9)	11.16 *	<0.001
Without ON tendency (<35)	367 (67.1)	436 (76.1)		
ORTO-15 score (M ± SD)	35.9 ± 4.1	37.2 ± 4.2	4.95 **	<0.001
‘Bottom body & weight’ *n* (%)				
Dissatisfaction	270 (49.4)	257 (44.9)	2.28 *	0.131
Satisfaction	277 (50.6)	316 (55.2)		
‘Upper body’ *n* (%)				
Dissatisfaction	269 (49.2)	275 (48.0)	0.16 *	0.692
Satisfaction	278 (50.8)	298 (52.0)		

* Chi squared test; ** Student’s *t*-test (t).

**Table 7 nutrients-11-00100-t007:** Orthorexia nervosa tendency and body satisfaction in the total sample.

Variables	ON Tendency (<40)	Without ON Tendency (<40)	Chi-Squared Test	*p*-Value	ON Tendency (<35)	Without ON Tendency (<35)	Chi-Squared Test	*p*-Value
	*N* = 840	*N* = 280			*N* = 317	*N* = 803		
‘Bottom body & weight’ *n* (%)								
Dissatisfaction	392 (46.7)	135 (48.2)	0.20	0.653	139 (43.9)	388 (48.3)	1.82	0.177
Satisfaction	448 (53.3)	145 (51.8)			171 (56.1)	415 (51.7)		
‘Upper body’ *n* (%)								
Dissatisfaction	392 (46.7)	152 (54.3)	4.88	0.027	133 (42.0)	411 (51.2)	7.75	0.005
Satisfaction	448 (53.3)	128 (45.7)			184 (58.0)	392 (48.8)		

**Table 8 nutrients-11-00100-t008:** Dietary patterns by orthorexia nervosa tendency and body satisfaction: adjusted logistic regression model (odds ratios with 95% confidence interval).

ON Tendency/Body Satisfaction	Dietary Patterns											
	High-Sugar Products & Snacks ^c^ (Ref. Bottom Tertile)		Fresh Products & Nuts ^c^ (Ref. Bottom Tertile)		Fatty Products & Dressings ^c^ (Ref. Bottom Tertile)		Oils & Potatoes ^c^ (Ref. Bottom Tertile)		Dairy Products & Whole-Meal Bread ^c^ (Ref. Bottom Tertile)		Meat ^c^ (Ref. Bottom Tertile)	
	Upper Tertile	*p*	Upper Tertile	*p*	Upper Tertile	*p*	Upper Tertile	*p*	Upper Tertile	*p*	Upper Tertile	*p*
Orthorexia nervosa ^a^ (ON tendency <40) ^b^	0.32 ^d^ (0.22; 0.46) ^e^	<0.001	2.30 (1.59; 3.32)	<0.001	0.59 (0.41; 0.85)	0.004	0.93 (0.65; 1.32)	0.690	1.06 (0.74; 1.51)	0.771	1.15 (0.79; 1.68)	0.471
Orthorexia nervosa ^a^ (ON tendency <35) ^b^	0.33 (0.24; 0.48)	<0.001	2.46 (1.72; 3.52)	<0.001	0.52 (0.36; 0.75)	<0.001	0.75 (0.53; 1.05)	0.094	0.93 (0.66; 1.32)	0.687	1.00 (0.70; 1.43)	0.994
‘Bottom body & weight’ ^a^ (dissatisfaction) ^b^	0.92 (0.66; 1.28)	0.606	0.82 (0.59; 1.15)	0.248	0.94 (0.66; 1.32)	0.714	0.81 (0.59; 1.13)	0.217	1.00 (0.72; 1.39)	0.992	0.78 (0.55; 1.11)	0.163
‘Upper body ’ ^a^ (dissatisfaction) ^b^	1.75 (1.29; 2.37)	<0.001	0.65 (0.48; 0.88)	0.005	1.03 (0.76; 1.41)	0.832	0.76 (0.57; 1.03)	0.073	1.02 (0.75; 1.37)	0.922	1.07 (0.78; 1.47)	0.669

^a^ dependent variable; ^b^ predicted level of dependent variable; ^c^ independent variables; ^d^ OR point estimate (e^β^) upper tertile vs. lower tertile; ^e^ 95% Wald Confidence Intervals. All data adjusted for sample weights. ORs were adjusted for: gender (categorical, female/men), age (continuous, years), place of residence (categorical, village/city ≤ 100.000 citizens/city > 100.000 citizens), level of studies (categorical, bachelor or engineering studies/master studies), college major (categorical, health related/non health related), BMI (continuous, kg/m^2^); *p* < 0.05 (Wald’s test).

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
