# Peer review of "Orthorexic Tendency in Polish Students: Exploring Association with Dietary Patterns, Body Satisfaction and Weight"

_nutrients, 2019, doi:10.3390/nu11010100_

Round 1
Reviewer 1 Report
This manuscript presents observational research among university students 18-35y from seven universities in Poland. It explores ON as measured by a validated tool, and explores it’s association with dietary patterns and body dissatisfaction measures.
Overall the manuscript is well written and presents confirmatory findings. It requires substantial revisions as follows:
Topic: The knowledge and clinical significance of ON is still emerging, so this work advances the knowledge on this topic.
Abstract:
Line 22 – indicate P values for difference in ON scores
Methods: Consent: ethics approval and informed consent contained, can the authors verify the students were not coerced into participating, given that the study was conducted by the teacher, during class time. Did students fear non-participating would effect grades?
The ORTHO-15 is not a diagnostic tool, so the authors should take care with any language used that implies it is a clinical diagnostic.
Questionnaire of Body Particular Parts and Parameters Satisfaction tool: It is unclear whether this tool is valid or reliable? I cannot locate the reference which is also in Polish. This is critical as it’s one of the key measures in your anaylsis, so this should be addressed by the authors.
The authors state they have used FFQ-6, has this been validated for use in the target population? Is it specific to the Polish food supply and nutrient composition of foods?
It is stated that the FFQ contained 165 products and dishes, and that the authors selected only 26 products and dishes. What was the justification for this? There is no theoretical or clinical basis for why the foods included in the PCA were selected, this should be clearly explained. A table can be helpful to illustrate the food group and examples of foods from the FFQ classified under the categories.
PCA Procedure: Why did the authors choose a cut off of 0.5 for their analysis? Usually a cut off of >0.4 is acceptable. Can you please show the other factor loadings in your table for the other foods.
Line 182-193 – this text repeats information that is presented in the table so should be removed, and direct the reader to the table in the text.
Self-reported weight: the authors should acknowledge limitations and bias in using self-reported body weight for BMI, especially the bias that may exist for the health-major students.
Results: Did the authors consider looking at Nutrition and Dietetics majors separately? There is literature that suggests that this group may be at higher risk for eating disorders, so it would be interesting to do a sensitivity analysis and see whether your results change if you analyze these as a separate group, or exclude them.
BMI should be treated as a continuous variable as a confounder in the anaylsis.
Results tables: please include a footnote for the table 8 to document the list of confounders included in the analysis.
Discussion: Three of the dietary patterns identified explained 5.4% of the variance or less in overall dietary patterns. The authors should elaborate on this limitation in the discussion.
What is the clinical significance of the difference in mean ORTO-5 scores observed between the health and non-health majors?
Author Response
Dear Reviewer,
I am sending an improved manuscript and responses to comments.
Kind regards,
Marta Plichta

Reviewer 2 Report
Orthorexia Nervosa is a controversial issue since clear and shared diagnostic criteria are missing.
In this paper, Plichta et.al. aim to assess the association of ON tendency with dietary patterns and body satisfaction among a large sample of polish students.
The paper is well-written and the methodology clearly explained. Results are partially in line with the previous literature, but limitations and strengths of the study are outlined.
Overall this paper add some insights into a still expanding and most unknown field
Author Response

(The authors gave the same response as above.)

Reviewer 3 Report
In the manuscript “Orthorexic tendency in students: an association with dietary patterns and body satisfaction”, Plichta et al. provided valuable information about orthorexic tendency in Polish students through a questionnaire survey among 1120 students and investigated the relationship between ON and dietary patterns and body satisfaction. Listed below are my comments and suggestions.
1. As the authors mentioned in the page 10 line 354, I think it is more precise and clear to specify “in Polish students” in the title.
2. In the table 1, because the sample age is from 18-35, so the age range should be clear and please use 18-20 and 25-35 instead of<20 and="">35.
3. Page 6, line209-219. How authors calculated the Odds ratios and confidence interval? This part only mentioned “Odds ratios (ORs) represented the chances of adherence to the upper tertiles of each DPs” but lacks the clear definition of ORs and detailed information about how ORs was calculated. Is there any reference for this part? As related, the table 8 also lacks necessary legend to make table self- explanatory, such as the setting of reference groups (OR = 1.00), the number of each group.
The design of 1st raw in table 8 is ambiguous and confusing, please revise.
4. Page 10 line 344, “analyzes” should be “analysis”
Line 347-352, this part of discussion is very redundant and not clear, please rewrite this part.
5. The paper “Prevalence of orthorexia nervosa is less than 1 %: data from a US sample (ref. 15)” discussed the limitation of ORTO-15. It is likely unable to distinguish between healthy eating and pathologically healthful eating. Do the authors have any data or information about the percentage of students who had severe healthy eating self-constraint and their diet had led to impairment in everyday activities and medical problems?
6. Ref. 65,75 format, line break.
Author Response

(The authors gave the same response as above.)
